# The Effect of Flow Rate on a Third-Generation Sub-Ohm Tank Electronic Nicotine Delivery System—Comparison of CORESTA Flow Rates to More Realistic Flow Rates

**DOI:** 10.3390/ijerph18147535

**Published:** 2021-07-15

**Authors:** Evan Floyd, Sara Greenlee, Toluwanimi Oni, Balaji Sadhasivam, Lurdes Queimado

**Affiliations:** 1Department of Occupational and Environmental Health, Hudson College of Public Health, University of Oklahoma Health Sciences Center, Oklahoma City, OK 73104, USA; Greenlee.sara@yahoo.com (S.G.); toluwanimi-oni@ouhsc.edu (T.O.); 2Department of Otolaryngology, College of Medicine, University of Oklahoma Health Sciences Center, Oklahoma City, OK 73104, USA; Balaji-Sadhasivam@ouhsc.edu (B.S.); Lurdes-Queimado@ouhsc.edu (L.Q.)

**Keywords:** electronic cigarettes, nicotine yield, flow rate, power settings, CORESTA, PMTAs

## Abstract

Many types of electronic cigarettes (ECs) are currently in use, but the default flow rate used to simulate puffing is centered on tobacco cigarette flow rates. CORESTA offers several methods and technical guides for evaluation of ECs but there are few puffing topography studies focusing on sub-ohm ECs; differences between real-world usage and that found in the literature appear large. This study focuses on how power and flow rate affect the nicotine yield of a sub-ohm EC. A puffing system (Puff3rd) has been designed and used to produce and collect EC aerosol. Nicotine yield was measured by GC–MS at three power levels and four flow rates. Data analysis was conducted in SAS using the MIXED procedure. Power, flow rate, and their interaction were all significant predictors of nicotine yield. Nicotine yield increased with both the vaping power and the puff flow rate with significant interaction of the two. Findings indicate that using the current CORESTA flow rate (1100 mL/min) to evaluate third-generation ECs underestimates nicotine yield and likely overestimates pyrolysis products. Real users are expected to have 2–3× the nicotine dose measured at 1100 mL/min, which could confound epidemiological studies seeking to link nicotine delivery to product satisfaction and acceptability.

## 1. Introduction

Electronic cigarettes (ECs) are widely accepted as a less harmful means of delivering nicotine than combustible tobacco products, especially cigarette smoking [1]. With the rise of the “youth vaping epidemic” [2], there is still much uncertainty about the role of ECs in tobacco control though they have been shown to be more effective in helping adults quit smoking than most other approaches [3,4,5,6]. In July 2019, a court ruling ordered the FDA to accelerate the premarket tobacco product application (PMTA) deadline for all electronic nicotine delivery systems (ENDS) products to May 2020, up from the original deadline of 2022 [7].

First- and second-generation ECs are usually puffed in a similar manner as combustible cigarettes, using a mouth-to-lung inhalation style. That is, the device is puffed with suction from the oral cavity, then the puff is inhaled from the mouth into the lungs [8,9]. With third-generation ECs, the inhalation style is typically direct lung inhalation [10], meaning the device is puffed with suction from the lungs (diaphragm) and that puff volumes are much larger. The Cooperation Centre for Scientific Research Relative to Tobacco (CORESTA) technical guide (CTG) for the Selection of Appropriate intense Vaping Regimes for E-Vapor Devices [11,12,13] provides a summary of topography studies conducted up through 2016. While many studies are available on first- and second-generation ECs, there were few that included third-generation ECs and those only measured puff duration [14]. Since there are no quantitative studies measuring the flow rates of third-generation users, the volumes and flow rates must be estimated.

ECs are battery-powered devices that create an aerosol to be inhaled by the user through a heated evaporation process. The atomizer of the EC contains a heating element in contact with a wick that connects to a reservoir. The reservoir is filled with an e-liquid solution that is typically a mixture of propylene glycol (PG), vegetable glycerin (glycerol) (VG), flavoring agents, water, and nicotine [15]. Currently, there is a large diversity in EC products that are used in a wide variety of manners. Most notably the power levels, puff volume and puff flow rates of third-generation ECs are much higher than first- and second-generation ECs as well as combustible cigarettes [16,17].

Korzun et al. [10] performed a study where the effect of airflow on aerosol consumption was measured. They selected 7.0, 18.3, and 36.0 mL/s airflows to study. They used the CORESTA airflow of 18.3 mL/s as the intermediate airflow. They also used 7.0 mL/s as it was the minimum airflow required to activate one of the first-generation devices, and 36.0 mL/s as it approximates a relatively high range reported in recent vaping topography investigations of tank-style (G2) atomizers. They found that a faster flow rate generally leads to less dense aerosols, because of the larger puff volume, but greater e-juice consumption due to more evaporation in the same puff time [10]. As Robinson et al. [18] so deftly stated, “unless the scientific community (and FDA) get ahead of the need to evaluate EC products in a realistic manner, we are likely to have irrelevant evaluation standards that must be overhauled at a later time, as was the case with tobacco cigarettes.” Korzun et al. [10] found that a modest increase in flow rate made a difference in the nicotine yield of a G2 atomizer. The functional difference in a G1 and G2 atomizer is much smaller than the difference in a G1 and G3 atomizer. We hypothesize that the effect of realistic flow rates for evaluating G3 atomizers will be even larger than what was observed by Korzun et al. [10]. The tidal volume of an adult at rest is approximately 0.5 L per breath. Let us assume that a puff is similar to a normal breath when using a direct lung inhalation technique. Using the puff duration of 3.2 s for mods (i.e., third-generation ECs) from St Helen et al.’s [19] topography study [20], we can estimate the puff flow rate as ~156 mL/s (~9.4 L/min), which is 2.5× the flow rate reported by Cunningham et al. [21] of 60.6 mL/s for tank EC users (i.e., second-generation ECs) and approximately 8.5× higher than the recommended flow rate from CRM No 81 [12].

In the PMTA, the FDA states that products should be evaluated in a manner as representative of real use as possible and requests that applicants evaluate their product for 31 harmful and potentially harmful compounds (HPHCs). The FDA specifically suggests evaluations with multiple e-liquid solutions at multiple nicotine levels be conducted since these HPHCs may be intrinsically present in the product or produced by the product under normal use or under intense use conditions. Unfortunately, the parameters that make up normal use and intense use are not specified; this is probably due to the diversity of ENDS products and incomplete topography literature describing every kind of product [22].

In this study, we investigated the effect of flow rate on nicotine yield of a third-generation EC at a variety of power settings, comparing the nicotine yield at the recommended flow rate of 1100 mL/min from the CRM No 81 [12] to higher values similar to Cunningham and even higher, more realistic flow rates. It should be noted that these higher flow rates do not represent the intense regime, rather the normal use condition. Some yet higher flow rates (and probably higher power) would represent an intense regime. This research focuses on the following hypotheses.

**Hypothesis** **1** **(H1).**
*Increasing vaping power will increase nicotine yield.*


**Hypothesis** **2** **(H2).**
*Nicotine yield will change with increasing puff flow rate.*


## 2. Materials and Methods

### 2.1. Experimental Design Overview

To evaluate the suitability of the cigarette-based evaluation methodology for third-generation ECs, nicotine yield was evaluated for a particular EC device: the Evolve DNA-75 research electronic cigarette (battery unit) coupled with the Tobeco Super Mini Atomizer, 0.5 ohm resistance coil. Nicotine yield of this EC setup was evaluated in triplicate for sets of 10 puffs at 3.2 s puff duration at low-, medium-, and high-power settings (25, 50 and 75 W) and at four flow rates (1100, 3000, 4500, and 6000 mL/min) (Table 1). These flows were chosen to range from the reference method flow rate of 1100 mL/min to well above the highest flow rate of ~3700 mL/min noted in CTG No 22 [14], observed by Robinson et al. [18]. Each trial condition in our study was conducted in triplicate to cover 3 power levels × 4 flow rates × 3 replicates = 36 trials. A randomized block experimental design was employed, with blocking on flow rate. A flow rate level was randomly selected, then the order of each power level condition and replicate was randomized within that flow rate block. Method blanks (MBs) were performed in a worst-case scenario after each 75 W trial. MBs were performed by puffing through the atomizer at the designated flow rate but without powering the atomizer. Results were MB corrected by subtracting MB values from powered trials with the same flow rate. The same coil and same atomizer were used throughout these experiments to reduce variability from different devices.

### 2.2. Puffing System and Sampling Train

An automated syringe-based puffing machine was constructed for use in this study. The puffing system was dubbed “Puff3rd” since it was designed to provide puff flow rates and volumes necessary for evaluating third-generation ECs under realistic use conditions. Briefly described, Puff3rd (Figure 1) uses a 3 L spirometry calibration syringe with the plunger position driven by a stepper motor. This system was designed to provide controlled puff rate, duration, volume, and puff replicates as well as push-button puff initiation. 

Puff3rd was calibrated by relating the number of steps performed by the stepper motor to the volume displaced by the syringe. This was accomplished by measuring internal dimensions of the syringe barrel to find cross sectional area, then multiplying cross sectional area by the linear distance traveled by the plunger when moved 10,000 steps to obtain a volume. The time to execute the 10,000 steps was used to convert puff volume to puff flow rate. After this initial calibration, a volumetric displacement calibration was conducted dispensing air from the syringe into an inverted graduated cylinder filled with water. As air entered the graduated cylinder the water was displaced, and the air volume read using the cylinder markings. This final calibration using a primary standard allowed fine-tuning of the instrument with minimal bias and high linearity (R^2^ > 0.999). After final calibration, 10 blind puff volumes ranging from 80 to 1900 mL were presented to a research assistant to determine the volume displaced. Each of these volumes were measured within 2.5% of the programmed value. Lastly, a programmable button pusher was included in the system to initiate puffs from the battery system. The timing accuracy of our button pusher should be +/−1 ms. This was verified through video recording and frame analysis to be within 0.05 s of the expected duration, which was limited by our video capture rate of 20 frames per second. Therefore, the volume and time were independently verified and found to be accurate to within 2.5%, and these parameters are combined to achieve a desired flow rate.

### 2.3. Aerosol Collection and Sample Preparation

EC aerosol was drawn by negative pressure (suction) through a short length of 0.375 inch ID tubing into bubblers (also known as gas washing bottles) with coarse fritted-glass diffusers and methanol as the collection solvent. The bubbler was followed by a short length of 0.375 inch ID tubing to a 47 mm filter cassette (Cat# 225-8496, SKC Inc., Eighty Four, PA, USA) containing a glass fiber filter (Cat# 09940028, Fisher Scientific, Hampton, NH) to capture particulate that may pass through the bubbler. This setup was selected after performing several methodological trials evaluating the capture efficiency of our bubblers and filters. In those preliminary trials, a series of 10 sequential replicate trials were conducted with two bubblers in series and a GF/B filter preceding or following the bubblers (BBF for bubbler-bubbler-filter and FBB for filter-bubbler-bubbler configurations). In the BBF configuration, 98.8% (+/−0.3%) of total nicotine mass was captured in the bubblers and 1.2% on the filter. In the FBB configuration, 98% (+/−0.8%) of total nicotine mass was captured on the GF/B filter and 2% in the gas washing bottles. This demonstrated that most of the nicotine was found in the particle phase and that both the filter and gas washing bottles were efficient at capturing most of the aerosolized nicotine mass. Since the filter alone was able to collect 98% of the aerosolized nicotine mass, we opted to simplify the sampling train and have a single bubbler followed by the filter. This allowed easier cleaning of components, ensured that the filter did not clog with particulate matter, and ensured well over 98% of nicotine was captured. For the 1100 mL/min flow conditions, a single fritted-glass midget bubbler (7532-20, Ace Glass Inc., Vineland, NJ, USA) was used, and for the 3000 mL/min flow, two midget bubblers were placed in parallel, as shown in Figure 1). For the two higher flow conditions, a 125 mL gas washing bottle was used (Cat#LG-3761-100, LabGlass Inc., Vineland, NJ, USA).

After each puffing trial was completed, bubbler contents were emptied into a graduated cylinder and the filter transferred to a glass VOA vial with PTFE lined lid. The bubbler, filter housing, all tubing, and forceps were rinsed 3× with methanol and the rinsate added to the graduated cylinder. A final volume was recorded, contents were transferred to the VOA vial and tumbled for 20 min to mix contents and extract the filter. Finished samples were stored at −20 °C until ready for analysis.

### 2.4. Puff Simulation

Trials were conducted in triplicate using a blocked randomized study design with each trial consisting of 10 standardized puffs executed with the Puff3rd puffing system. Puff3rd can comply with CRM No. 81 [12] prescriptions, but this system was designed for evaluation of ECs that are used at much higher flow rates and larger puff volumes. Certain adaptations were applied to enable conducting an intense puffing regime, but this evaluation represents normal use of these ECs rather than intense use. Puff flow rate, duration, volume, number, and frequency were controlled via the Puff3rd program and are represented in Table 1. The Puff3rd syringe was connected to the atomizer through the sampling train described above. Puff time was fixed at 3.2 s after St. Helen et al. [19] and CORESTA No. 22 [14] description. Since puff time was fixed but puff flow rate was varied, puff volume also varied across flow rate conditions. Four puff flow rates were evaluated at three power settings and compared against the values obtained from method blanks performed at the same flow rate. Power was delivered to the atomizer by the Evolve EC battery unit, which was maintained above 50% charge. Each trial consisted of 10 puffs conducted on a 30 s cycle. For all trials, the same lab-prepared e-liquid was used. Approximately 100 mL of e-liquid was prepared to be 70% volume VG, 30% volume PG and 12 mg/mL nicotine (Sigma Aldrich, Cerilliant N-048-1ML, Round Rock, TX, USA). Aliquots were prepared and frozen at −20 °C, fresh aliquots were used daily.

The EC atomizer was weighed before and after each 10-puff trial to quantify e-juice mass vaporized during the trial. A flow rate was randomly selected, programmed into Puff3rd and verified with a flow meter (Bios DryCal, Defender 510, MesaLabs, Lakewood, CO, USA). Puff duration was controlled by the button pusher depressing the “fire” button of the battery unit for 3200 ms. Puff flow was programmed for 4.0 s with a 0.3 s delay before initiating the button pusher to allow steady flow in the bubbler. Puff flow was continued for 0.5 s after the atomizer turned off to clear the tubing of residual aerosol. After the trial was completed, the sample was transferred to a VOA vial and the sampling train cleaned as described above.

### 2.5. Sample Analysis

Samples were analyzed for nicotine content by GC–MS using an internal standard correction technique. A small aliquot of each sample was transferred to a 1.5 mL GC–MS vial for analysis. The GC–MS was calibrated for nicotine from 0.1 to 25 µg/mL using the 87 *m*/*z* mass fragment for quantification with 134, 160 and 161 *m*/*z* fragments for qualification. The found nicotine concentration and sample volumes were used to calculate the total mass of nicotine vaporized and divided by number of puffs per trial to get nicotine yield per puff.

The instrument used for these analyses was an Agilent 6890 with 5973N MSD (Agilent, Santa Clara, CA, USA). GC parameters were as follows: inlet 270 °C, 0.32 mm i.d., 30 m DB-5 column, column flow rate 1.5 mL/min, constant flow, 1 µL injection, oven start 70 °C, ramp rate 15 °C/min to 170 °C, hold 0.5 min, ramp rate 40 °C/min to 270 °C, hold 2 min. MS parameters were as follows: interface 270 °C, source 230 °C, EI mode, 70 EV, quadrapole 150 °C, scan mode 30–280 *m*/*z*. The methanol used in all aspects of this experiment (bubbler solution and rinse solvent) was prepared at 10 ppm internal standard to correct for evaporation and injection variability. Dichlorobenzene was used as the internal standard due to its low volatility and absence from the sample matrix.

### 2.6. Data Analysis

Plots were constructed with nicotine yield per puff versus atomizer power (Figure 2) and nicotine per puff versus flow rate (Figure 3) to visualize the trends and interactions of the main effects. Plots were fit with functions available in Excel to find the best fit, with quadratic providing the highest correlation coefficients and least biased residuals. Similar plots were made for the mass vaporized per puff data (Figure 4 and Figure 5). Statistical comparisons were made with a mixed-effects multivariate regression using SAS 9.2 (SAS Institute Inc., Carry, NC, USA) with power and flow rate as main effects and power*flow as the interaction term. The mixed-effects model was selected for its robustness and tolerance for non-normality of data. Significance was set at alpha ≤ 0.05. 

## 3. Results

A summary of nicotine yield per puff and vapor power is shown in Table 2; nicotine yield per puff, mass vaporized per puff and nicotine yield normalized by mass vaporized. To visualize this data, each main effect was examined without considering the interaction term. Figure 2 reflects the effect of power on nicotine yield at different flow rates. It is clear that increasing power increased nicotine yield, though not in a linear manner and with differences across flowrates (reflecting the interaction). This was expected to be a linear relationship under the assumption that changes in power and vaporization would be proportional [23]; however, a linear fit of these data resulted in high residuals with over-prediction at 25 and 75 W and under-prediction at 50 W. The second-order polynomial function had well balanced residuals and fit the data much better than a linear fit, with R^2^ values between 0.944 and 0.992. This experimental dataset has been published to the Harvard Dataverse and is freely available for download and review.

It is also evident that increasing flow rate increased nicotine yield since each series is distinctly higher than the previous. However, the data are compressed, especially at the 25 W setting, so it is easier to see these differences by re-examining with emphasis placed on flow rate, as shown in Figure 3; the *x*-axis is now flow rate and the series are power levels. In Figure 3, it is apparent that nicotine yield increases with flow rate at all power levels, but the greatest effect is in the 75 W series. At 1100 mL/min flow rate, the nicotine yield at 75 W is only 19% larger than at 50 W, but at 6000 mL/min, the nicotine yield is 44% larger, which is near to the expected 50% increase expected based on thermodynamics. That is, 75 W will deliver 50% more energy than 50 W during the 3.2 s puff and should result in 50% more e-liquid vaporization. Visually, it is clear that these series begin to diverge to a greater extent as flow rate increases, but a more subtle aspect is the plateauing of the 50 W series, whereas the 75 W series continues to increase in a near linear relationship. For all trials, nicotine yield increased with flow rate; the 25 W series increased 89%, from 64 to 121 µg/puff; the 50 W series increased 112%, from 163 to 346 µg/puff; and the 75 W series increased 156%, from 195 to 499 µg/puff. Nicotine yield increased markedly as flow rate increased and became nearly linear at the highest flow rate (6000 mL/min).

The mass of e-liquid vaporized (mass loss per puff) was also tracked for each trial and the results are shown in Figure 4. Interestingly, the difference between the CRM 81 [12] flow rate (1100 mL/min) and all the higher flow rates is quite large, but the differences among the higher flow rates are not large. Also interesting were the method blank results; MBs for nicotine were all found to be slightly positive, but the mass loss values were negative (meaning the atomizer gained weight during the MB trials, most likely from absorption of water vapor), but a slight vaporization of nicotine occurred.

While these graphs help to visualize the relationship between nicotine yield and mass loss at different flow rates, they fail to account for the potential interaction between these two main effects. In this case, a significant interaction is expected. Simply heating the atomizer would not generate much aerosol. Similarly, puffing without heating the atomizer would not generate much aerosol. However, when puffing and heating are performed at the same time, aerosol is clearly generated, thus it is the interaction of puff flow and power that is expected to generate aerosol. Results from the mixed-effects multivariate regression showed that both main effect and the interaction had significant effect on nicotine yield per puff. All *p*-values were <0.0001 with 24 degrees of freedom.

## 4. Discussion

### 4.1. Nicotine Vaporization

Talih et al. [23] reported that increasing the puff duration (2, 4 and 8 s) and increasing voltage (3.3 and 5.2 V) resulted in higher nicotine delivery. Currently, there are no studies available on the effect of flow rate and vaping power on nicotine yield. In the present study, we have observed a significant increase in the nicotine yield when vaping power increased and when flow rate increased. The nicotine yield does not remain proportional to the bulk solution with increases in either power or flow rate. This shows that the nicotine yield will depend on both the flow rate and the power, and as one topography factor is changed it will affect the other. Qualitatively, the relationship between vaping power and nicotine yield at each flow rate was not linear as was expected; rather, it plateaued at lower flow rates and was reasonably well described by the second-order polynomial (quadratic) model (R^2^ > 0.94–0.99). 

The plateauing observed within a given flow rate (Figure 2) indicates there is a factor acting against mass transfer (vaporization) such as the wicking rate of the atomizer being unable to supply enough e-liquid to match the increasing power and thus limits the nicotine yield [24]. Another factor that could reduce nicotine yield is re-deposition of e-liquid vapor back onto the wick or inside of the atomizer [23]. It is possible that the kinetics of redeposition by molecular diffusion begin to match pace with vaporization of e-liquid when the vapor concentration reaches super saturation conditions that are certainly expected within the atomizer [17,25]. At high power and lower puff flow rates, vapor could condense and redeposit within the atomizer and be resorbed by the wick. Slower flow rates would result in higher vapor concentration and longer residence time in the atomizer, each of these would facilitate increased deposition within the atomizer due to Brownian motion [25].

The fact that the plateauing is largely ameliorated by increasing flow rate indicates that wick supply is not the limiting factor. If supply of e-liquid was the limiting factor, then increasing flow would have minimal effect. However, since flow shows a large effect on nicotine yield, it is more likely that re-condensation and re-deposition within the atomizer are the dominant causes of reduced nicotine yield at low flow rates.

### 4.2. Total Mass Vaporization

The present study observed a significant increase in vaporized mass per puff when vaping power was increased, which is consistent with earlier reports [10,17]. Additionally, the present study observed a significant increase in vaporized mass per puff as flow rate was increased (Figure 4). 

For each puff flow rate, the relationship between vaping power and total mass vaporized is not linear; the increase in total mass vaporized plateaus markedly at 1100 mL/min even though power is increasing, while vaporization rates are larger, more linear and more similar to each other at the three higher flow rates tested. Since the mass vaporization curves do not parallel the nicotine yield curves, we must conclude that nicotine is not vaporized at the same efficiency under different flow conditions and that some other component of the mixture is vaporizing more readily at higher flow rates. This is most likely due to differences in the boiling points of the e-liquid components (PG, VG, and nicotine). PG has a boiling point of 188 °C, which is substantially lower than that of nicotine (247 °C) and VG (290 °C) [26]. The estimated boiling point of a 30:70 PG:VG solution is 225 °C and with 3% mole fraction nicotine 215 °C [27]. The data shown here indicate all components are at saturation vapor pressure at 1100 mL/min flow rate as evidenced by plateauing in the mass vaporized with increasing power. Increasing flow rate to 3000 mL/min caused at least one component of the solution to begin evaporating more freely as evidenced by the sharp rise in mass vaporization at the elevated flow rates. Since PG has the lowest boiling point, it is most likely additional PG vaporization that accounts for the higher mass vaporization at elevated flow rates, but PG and VG concentration were not examined quantitatively in this study due to their very high concentrations and co-elution with the chosen GC parameters, therefore this point cannot be explored further.

If air were not limited (or vapor at saturation), then we would expect mass vaporization and nicotine yields to increase linearly with flow rate since higher flow rates provide more air volume to host the e-liquid vapor. Clearly, this was not observed (Figure 5), with the increase ranging from 1.9 to 2.5× instead of the expected 5.5×. This could be due to the cooling effect of the additional air flow and further highlights the need to evaluate G3 ECs under a variety of realistic use conditions.

### 4.3. Nicotine/e-Juice Mass Vaporization Ratio

CORESTA Technical Guide 22 [14] recommends normalizing nicotine yield by the mass of e-juice vaporized to reduce uncertainty in replicate trials. For each power level, the nicotine to total mass vaporization ratio increases with increasing puff flow rate but drops with increasing vaping power. The CORESTA recommendation seems to be useful for correcting errors caused by inconsistent device performance, but only when the amount of air is well above that needed for full vaporization or when only one flow rate is used [14]. If the vapor dynamics were static across flow rate or power level, we would have expected a constant ratio; however, as discussed above there seems to be a limitation in vaporization. In this study, we observed a suppression of nicotine yield at higher vaping power and lower flow rates. This indicates the suppression is a saturation effect. 

When air is limited, the vapor concentration reaches saturation quickly, and this flattens the diffusion gradient between the wick and puff air, which causes aerosol to form through condensation. With minimal diffusion gradient, the vaporized e-liquid has little motivation to move away from the wick surface except via Brownian motion. Vaporized e-liquid is expected to begin re-depositing on the wick and internal atomizer surfaces at a rate approaching to the vaporization rate. In an air-limited environment, the molecules most affected are those with lower saturation concentration, i.e., those with higher boiling points. When the air flow rate is low and the power is low, the air is nearly saturated but still has enough capacity for most of the e-liquid mass to be evaporated; at low flow and high power, the air is fully saturated, and the higher boiling point molecules do not diffuse away from the wick as quickly. Since nicotine and VG have boiling points much higher than PG, their vaporization is expected to be more suppressed than PG. At the highest flow rate, there is ~6× more air to host the vaporized e-juice and the residence time inside the atomizer is shorter by 6× due to the higher air velocity. Therefore, a process such as diffusive re-deposition would occur less with high flow than at a low flow rate.

### 4.4. Potential Effect on HPHCs

Since the typical operational flow rate of a G3 EC is expected to be 6 L/min, or greater, we will briefly explore implications of high real-world flow rates on HPHCs such as carbonyls. Other studies such as Beauval et al. [28] reported that modification in puffing conditions leads to significant variations in the carbonyl composition of e-cigarette aerosols. Havel et al. [29] demonstrated that increased voltage increases aerosol formation and showed significant voltage-dependent increases in aldehydes. This is consistent with Talih et al. [23] and Gillman et al. [24]. Increasing flow rate decreases the residence time within the atomizer, provides more dilution air (volume) to host e-liquid vapor, and has a cooling effect on the atomizer. Each of these factors are expected to reduce the kinetics of carbonyl formation through pyrolysis. If G3 ECs are evaluated in labs using unrealistically low flow rates, then we anticipate overestimation of the carbonyl content and underestimation of the nicotine delivery.

## 5. Conclusions

Nicotine yield increased by 3.1–4.1× with atomizer power and 1.9–2.5× with puff flow rate, but not linearly for either. The interaction of power and flow rate was strong and statistically significant, which means nicotine yields cannot be extrapolated from a known power-flow rate condition without also having the interaction effect characterized for the atomizer in question. As suggested by CTG No. 22 [14] and FDA guidance on PMTAs [22], ENDS should be evaluated under a range of reasonably expected use conditions and intense use conditions, including flow rates representative of real-world use rather than flow rates based off combustible cigarette puffing regimens. In the puffing topography literature, puff flow rates near 4 L/min have been observed, but higher values are expected to occur on a regular basis within the G3 EC category. Further evaluation is needed to determine the impact of puff flow rate on HPHC formation as this is expected to be overestimated for G3 ECs when tested at the CORESTA No. 81 [12] flow rate of 1100 mL/min. This has direct regulatory impacts as the FDA evaluates PMTAs submitted for G3 products and those with similar puffing patterns. If the standardized product characterizations are conducted at the reference flow rate of 1100 mL/min, then the nicotine yield is likely to be underestimated and HCHPs such as carbonyls are likely to be overestimated.

Additionally, further research is needed to better characterize the puff flow rates of G3 EC users. These studies must also be aware of the flow resistance introduced by the topography device since typical flow rates may be as high as 30 L/min and possibly even greater under intense use. A standardized method using higher (realistic) flow rates should be established for evaluating G3 devices to better reflect real use conditions. 

## Figures and Tables

**Figure 1 ijerph-18-07535-f001:**
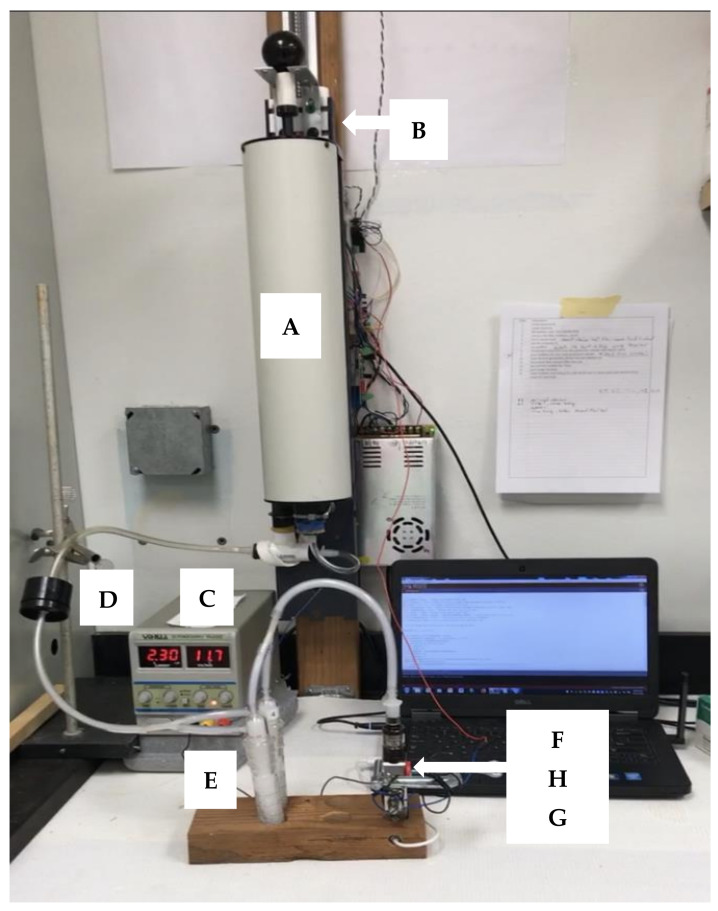
The PUFF3rd puffing machine setup with parallel midget bubblers for the intermediate flow rate of 3000 mL/min. Higher flow conditions (4500 and 6000 mL/min) used a 125 mL gas washing bottle. The lower condition (1100 mL/min) used a single midget bubbler. (**A**) 3 L spirometer syringe, (**B**) stepper motor, (**C**) adjustable DC power supply, (**D**) filter holder, (**E**) bubbler(s), (**F**) atomizer, (**G**) battery system and (**H**) clamp-on button pusher.

**Figure 2 ijerph-18-07535-f002:**
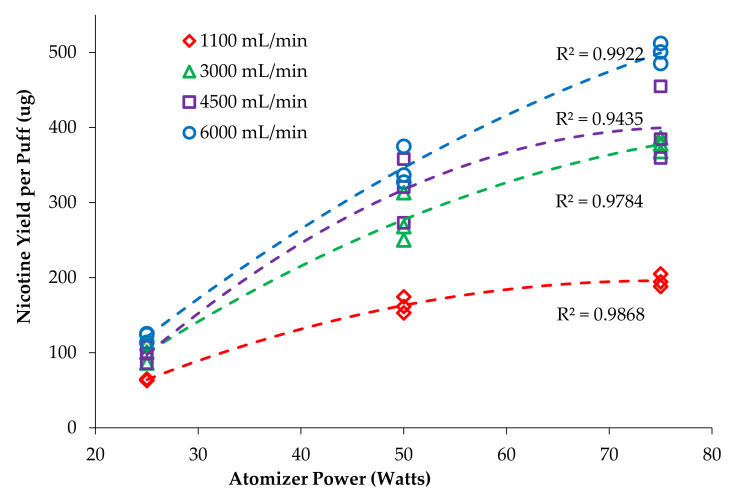
Effect of power on nicotine yield at different flow rates.

**Figure 3 ijerph-18-07535-f003:**
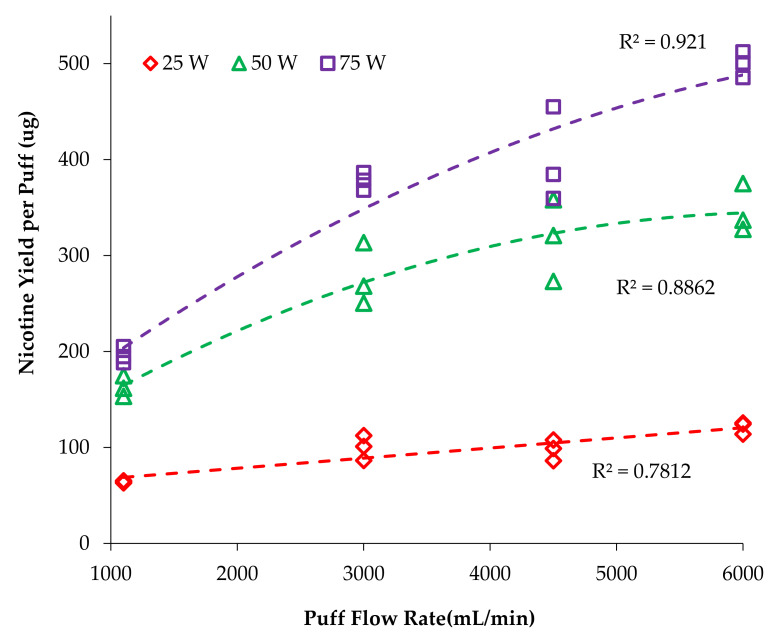
Effect of flow rate on nicotine yield at different power settings.

**Figure 4 ijerph-18-07535-f004:**
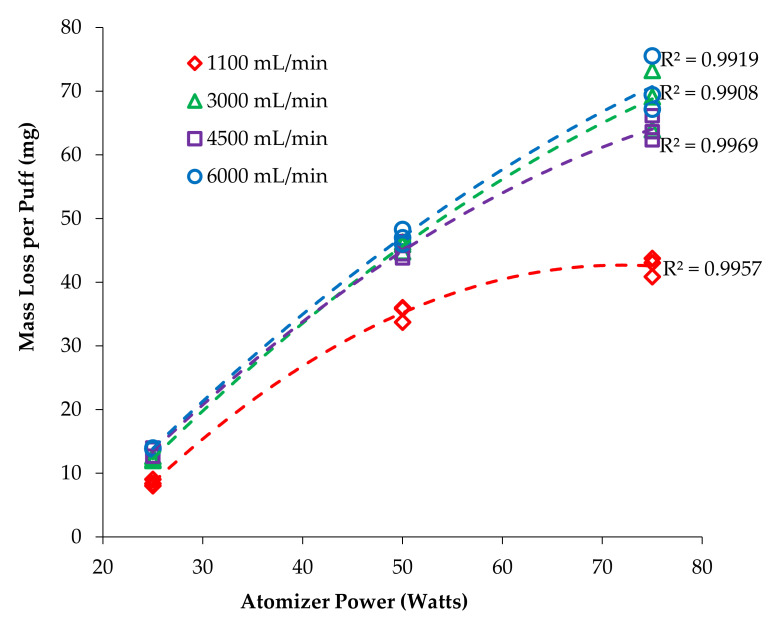
Effect of power on mass loss per puff for each trial at different flow rates.

**Figure 5 ijerph-18-07535-f005:**
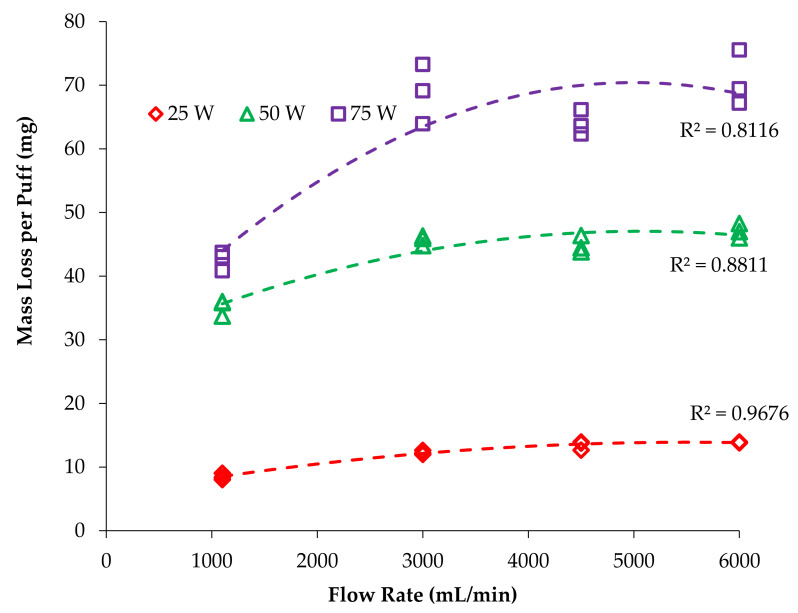
Effect of flow rate on mass loss per puff at different power settings.

**Table 1 ijerph-18-07535-t001:** Puff3rd Program Parameters.

Puff Parameters	Values
Flow Rate	1100–6000 mL/min
Duration	3.2 s
Volume	58.7–320 mL
Number	10 puffs
Frequency	1 puff/30 s

**Table 2 ijerph-18-07535-t002:** Summary of nicotine yield per puff and vapor power.

	Flow Rate (mL/min)	Nic (ug/puff)	CV	RSE	CI95% (lower:Upper)	ML (mg/puff)	CV	RSE	CI95% (Lower:Upper)	Nic/ML (ug/mg)	CV	RSE	CI95% (lower:Upper)
25 W Power:	1100	66.6	1.6%	0.9%	65.4; 67.8	8.48	6.0%	3.5%	7.9; 9.1	8.03	5.3%	3.0%	7.5; 8.5
	3000	99.8	13.0%	7.5%	85.2; 114.5	12.71	5.9%	3.4%	11.9: 13.6	7.85	7.4%	4.3%	7.2; 8.5
	4500	97.5	11.2%	6.4%	85.2; 109.9	13.47	5.1%	0.03	12.7; 14.3	7.24	15.7%	9.1%	6.0; 8.5
	6000	121.2	5.2%	3.0%	114.0; 128.4	13.89	0.5%	0.3%	13.9; 14.1	8.72	5.8%	3.4%	8.1; 9.3
50 W Power:	1100	165.6	6.5%	3.8%	153.4; 177.8	35.19	3.6%	2.1%	33.7; 36.6	4.79	3.9%	2.3%	4.6; 5.0
	3000	277.2	11.7%	6.8%	240.4; 314.0	45.68	1.7%	0.9%	44.8: 46.6	6.07	10.3%	6.0%	5.4; 6.8
	4500	317.2	13.4%	7.7%	269.0; 365.4	44.88	3.0%	1.7%	43.4; 46.4	7.07	13.0%	7.5%	6.0; 8.1
	6000	346.3	7.3%	4.2%	317.8; 374.9	47.09	2.4%	1.4%	45.8; 48.4	7.36	5.5%	3.2%	6.9; 7.8
75 W Power:	1100	198.4	4.3%	2.5%	188.8; 207.9	45.67	6.1%	3.5%	42.5;48.8	4.45	9.1%	5.3%	4.0; 4.9
	3000	377.5	2.4%	1.4%	367.1; 388.0	68.78	6.8%	3.9%	63.5; 74.1	5.49	5.6%	3.2%	5.1; 5.8
	4500	399.6	12.4%	7.2%	343.5; 455.6	64.04	3.0%	1.7%	61.9; 66.2	6.24	9.9%	5.7%	5.5; 6.0
	6000	499.4	2.7%	1.6%	484.2; 514.6	70.72	6.1%	3.5%	65.8; 75.6	7.06	7.2%	4.2%	6.5; 7.6

Nic = average nicotine yield per puff (ug/puff). CV = coefficient of variation, standard deviation expressed as a % of mean value. RSE = relative standard error, standard error expressed as a % of mean value. CI95% = 95% confidence interval of the mean. ML = mass of e-juice vaporized per puff, expressed as mg/puff. Nic/ML = nicotine yield/e-juice vaporized, expressed as ug/mg.

## Data Availability

The data presented in this study are available through the Harvard Dataverse at Floyd, Evan, 2021, “Puff flow rate affects nicotine yield of sub-ohm ENDS”, https://doi.org/10.7910/DVN/3FLZBW, Harvard Dataverse, V1, UNF:6:91H5fWo6o5SvAYx8PIYufA== [fileUNF].

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
