# Peer review of "The Effect of Flow Rate on a Third-Generation Sub-Ohm Tank Electronic Nicotine Delivery System—Comparison of CORESTA Flow Rates to More Realistic Flow Rates"

_ijerph, 2021, doi:10.3390/ijerph18147535_

Round 1

Reviewer 1 Report

I think overall its a good paper worth to publish but some minor comments should be addressed prior to publication.

Please see my comments and questions below and attached:

  • P1 L29: Electronic cigarettes (EC) are widely accepted as a less harmful means of delivering nicotine than combustible tobacco products, especially cigarette smoking [1].

Not sure that this is true, and reference is dated 2014. In particular due to EVALI crisis, I don’t think that e-cigs relative safety is widely accepted.

  • P11 L304-307 At high power and lower puff flow rates vapor could condense and redeposit within the atomizer and be resorbed by the wick. Slower flow rates would result in higher vapor concentration and longer residence time in the atomizer which would both facilitate increased deposition within the atomizer due to Brownian motion.

Yes, at high power and low flow rate residence time will increase and particles will grow to the larger (above micron size). Those particles will be more efficiently depositing inside the atomizer due to impaction forces but not Brownian motion. Brownian motion mostly affects particles below 50 nm size.

  • Regarding experimental setup design:

Filter after bubbler is not a very appropriate design. Normally filter should go first to catch particles and then bubbler to catch the gas phase, otherwise filter gets wetted by the moisture coming from the bubbler and it affects filter performance. You performed tests in both ways and results showed that both work good for nicotine collection, and it seems like its ok for sub-ohm devices that generate large size particles. Might not work that well for less powerful devices where particles are smaller.

Also, for data analysis if you have measured aerosol mass collected on the filter and nicotine collected on the filter then you can make straightforward ratio nicotine/aerosol mass. Instead, you reported this ratio based on weighing the atomizer before and after 10 puffs, and it sometimes works but possible wall losses on delivery tubing could affect accuracy of this method. Indirect indication of some inconsistency could be found in Table 2, where nicotine per puff is basically continuously increasing with both power and flow rate (only at 25 W there is a small drop from 3000 to 45000 ml/min), whereas mass of e-juice dependence upon flow rate is not that consistent (at 50 W there is a drop from 4500 to 6000 ml/min, at 75 W there is a drop from 3000 to 45000 ml/min).

  • Results

Decrease of nicotine per puff at 25 W when flow rate increased from 3000 to 4500 ml/min whereas at the same time mass of vaporized e-juice increased seems strange and very likely could be caused by nicotine sampling artefacts.

Decrease of vaporized e-juice mass while increasing flow rate at 50 W (from 4500 to 6000 ml/min) and 75 W (from 3000 to 4500 ml/min) while nicotine per puff was increasing probably indicate inaccuracy of the e-juice mass measurements.

Preferred standard method would be to compare nicotine mass vs so-called TPM (total particulate matter) collected on the filter.

  • I would recommend putting labels on Figure 1 to indicate different parts of the setup, syringe, bubblers, filter, e-cigarette.
  • Not sure what is the purpose of the paragraph: 4.4. Potential Effect on HPHCs.

You didn’t measure neither carbonyls nor metals. And these assumptions are not very relevant to the general purpose of this manuscript.

  • What was the content of e-liquid? PG/VG 30/70? What was the nicotine concentration?

Was it fresh solution or it was prepared and then stored, if second for how long it was stored and under what conditions (temperature first of all)?

Reviewer 2 Report

Manuscript ID: ijerph-1273222

The Effect of Flow Rate on a Third Generation Sub-Ohm Tank Electronic Nicotine Delivery System - Comparison of CORESTA Flow Rates to more Realistic Flow Rates

Evan L Floyd * , Sara Greenlee , Toluwanimi M Oni , Balaji Sadhasivam , Lurdes Queimado

     This article by Floyd et al addresses an important gap in modern e-cig research.  While many device characteristics are emphasized in other research, few studies have investigated the effects of higher flow rates used in 3rd generation devices. This study effectively compared high flow rates used in these products to that of a current standardized puffing profile. I recommend this manuscript for publication with revisions.

Major Revisions:

  1. Please make sure to include the ohm of the coil used, or the amperage of the device used or both. With only wattage we do not get a complete picture of the device used. This might fit around line 102.
  2. Method blanks were discussed in Lines 109-111 but not explained what they were here. However, they were described in lines 186-188. Please either move description from 186-188 to 109-111 or move text from 109-11 to 186-188.
  3. Figure 1 legend; For the gas wash collection, sometimes one gas washing bottle was used, sometimes two were used. In addition to this different volume gas washing bottles were used between flow rates. Do you think this effected the outcomes of the study? Why were different volumes and number of gas washing bottles used between exposures?
  4. Lines 159-161; the method describes using a single gas washing bottle (aka bubbler?) but the figure ledge describes using two for the 3000ml/min flow rate. Please make sure this is consistent. Is a bubbler the same thing as a gas washing bottle?
  5. Methods, calibration of homemade smoking machine, Lines 129-132; What temperature was the water and the room while preforming these calibrations? Were they done the same day? Do you think temp could have a significant effect on the calibration of the system? For example, a 5˚C temp change can result in a ±6mmHg change in pressure. If the experiment was not performed in the same conditions an effect on flow rate could be observed.
  6. Lines 151-161; Tests for number of bubblers was performed and determined that only one bubbler was needed based on using two bubblers and a filter in various combinations. Was only a single bubbler tested to verify collection efficiently as well? If so, what were the results of this trial?
  7. Lines 145-173; What solvent was used in the bubblers? Line 170 described use of methanol to rinse the parts in contact with aerosol but does not list what solvent was used. Lines 214-15 does mention methanol is used in all aspects of equipment. Please move or add to lines 145-173.
  8. Did eliquid ever form droplets inside of the tubing connected to the bubblers? This typically happens with 3rd generation devices using high flow rates. If this did happen would this be counted toward particulate phase results or excluded from the results? Was the tubing used anti- static to prevent this?
  9. Section 2.3; What concentration of nicotine, PG and VG were used? Was this lab made eliquid or commercial purchased? Where nicotine salts used? Was anything else added?
  10. Was the mAH of the battery recorded before and after experimentation? With the higher wattages like 50/75W the draw can have an effect on the consistent distribution of the aerosol depending on device. Do you think that this could have had an effect on the response curves not being linear?
  11. Lines 269-272; For the method blank, it was stated that slight vaporization occurred, so the 0≠ Do you think this occurred because of the flow rate or because the device malfunctioned and did not power off the device? Did you measure the wattage for the 25,50 and 75 to verify that they were delivering the expected wattage? I believe this could have compounding effects as all the data is normalized to this.
  12. What coil was used in the Tobeco Super Mini Atomizer? These tanks seem to come with multiple sub-ohm options. Was the same ohm coil used for all tests? If not how often was this coil changed?
  13. Were multiple Tobeco Super Mini Atomizer used in this study or just one that was continuously refilled? If the later did you see accumulation of burnt solvent on the coil? More descriptions of the device characteristics would help answer a lot of questions.

Minor Revisions:

  1. I believe CORESTA 81 (2015, Ref 12) uses a flow rate of 1110ml/min 18.6ml/s vs CORESTA 81 (2017, Ref 13) uses a flow rate of 1100ml/min 18.3ml/s. Please make sure in indicate the correct reference throughout the text rather than use both citations.
    1. Lines: 59,89
  2. Please change “flor” to “flow”, line 90
  3. Extra space before “Significance”, line 226
  4. Brackets needed around reference “12”, line 267
  5. Remove “–“ before the, line 291. I believe you were trying to place a semicolon here.
  6. You may want to reference “Table 1” in Lines 101-103.
  7. Please note in either figure legends of Figure 2-5 or in the results what function was used for each graph, linear or quadratic. I believe it was quadratic for all, however the discussion mentions that the functions were not linear a few times. Alternatively, you could add figure labels next to each graph type mentioned in Lines 219-227. For example, “nicotine yield per puff versus atomizer power, Table 1” Do you think there would be a benefit to showing the linear functions in supplemental?
  8. Line 255 “based on thermodynamics”; Is this referring to a specific thermodynamic law? What data supports this expected 50% increase? Citation?
